chemical engineering/materials science

recycling, hydrocracking, catalysis, zeolites, sustainability

**Author for correspondence:**
Aleksander A. Tedstone
e-mail: aleksander.tedstone@manchester.ac.uk

This article has been edited by the Royal Society of Chemistry, including the commissioning, peer review process and editorial aspects up to the point of acceptance.

# Transition metal chalcogenide bifunctional catalysts for chemical recycling by plastic hydrocracking: a single-source precursor approach

Aleksander A. Tedstone[1], Abdulrahman Bin Jumah[2], Edidiong Asuquo[1] and Arthur A. Garforth[1]

[1]Department of Chemical Engineering and Analytical Science, University of Manchester, Oxford Road, Greater Manchester, M1 3BB, UK
[2]College of Engineering, King Saud University, PO Box 800, Riyadh 11421, Saudi Arabia

(iD) AAT, 0000-0003-0152-8248; EA, 0000-0002-0530-0497

Sulfided nickel, an established hydrocracking and hydrotreating catalyst for hydrocarbon refining, was synthesized on porous aluminosilicate supports for the hydrocracking of mixed polyolefin waste. Zeolite beta, zeolite 13X, MCM41 and an amorphous silica-alumina catalyst support were impregnated with the single-source precursor (SSP) nickel (II) ethylxanthate for catalyst support screening. Application of this synthesis method to beta-supported nickel (Ni@Beta), as an alternative to wet impregnation using aqueous nickel (II) nitrate, provided catalytic materials with higher conversion to fluid products at the same mild batch reaction conditions of 330°C with appropriate agitation and 20 bar $H_2$ pressure. Mass balance quantification demonstrated that SSP-derived 5wt%Ni@Beta yielded a greater than 95 wt% conversion of a mixed polyolefin feed to fluid products, compared with 39.8 wt% conversion in the case of 5wt%Ni@Beta prepared by wet impregnation. Liquid and gas products were quantitatively analysed by gas chromatography–flame ionization detection (GC-FID) and gas chromatography–mass spectrometry (GC-MS), revealing a strong selectivity to saturated $C_4$ (37.3 wt%), $C_5$ (21.6 wt%) and $C_6$ (12.8 wt%) hydrocarbons in the case of the SSP-derived catalyst.

# 1. Introduction

Exploiting polyolefin materials to create a circular economy can be made possible via catalytic chemical recycling, as they represent a highly refined hydrocarbon source with similarities to fossil feedstocks, making chemical recycling an important supplement to mechanical recycling of plastics [1]. While some polymers such as poly(ethylene terephthalate) (PET) are amenable to repeated mechanical recycling, polyolefins degrade during reprocessing and are not suitable for many applications (such as food contact) after the temperatures and conditions of this classical approach to recycling. Polypropylene (PP), low-density polyethylene (LDPE), linear low-density polyethylene (LLDPE), medium-density polyethylene (MDPE), high-density polyethylene (HDPE) and polystyrene (PS) account for 55.4% of European plastics demand combined. If current trends continue, by 2050, there will be 12 000 Mt of plastic waste in landfills or in the natural environment globally [2], representing a vast pool of this hydrocarbon feedstock that will persist in the world until a solution is found.

Chemical recycling of polymers by hydrocracking, a catalytic refining process, offers the potential for the selective recovery of useful chemical fractions of the desired boiling range at relatively modest reaction conditions compared with pyrolysis processes. Hydrogen pressure is used to remove sulfur and chlorine impurities to yield products in the range of heavy diesel to light naphtha [3], with high iso-paraffin and low olefin content [4–6]. Hydrocracking requires a bifunctional catalyst with the acidic function enhancing the cracking typically provided by a high surface area support, such as a zeolite [7–9]. The hydrogenation/dehydrogenation function is provided by noble metals, such as Pt or Pd, and in a milder form of hydrocracking, known as hydrotreating, by transition metals using alumina support loaded with Mo, Co, W or Ni [10].

Bifunctional zeolite catalysts have been demonstrated to depolymerize polyolefins via hydrocracking [11] and herein the repertoire of available catalysts is expanded, by replacing expensive and earth-scarce precious metals, such as platinum and palladium, with earth-abundant transition metals and metal chalcogenides. Nickel sulfide, a common hydrocracking and hydrotreating catalyst for petroleum refining, can be synthesized from the single-source precursor (SSP) nickel (II) ethyl xanthate [12]. Using this versatile method to impregnate acidic supports is a new application of an established synthesis method. Bifunctional catalysts are created traditionally by wet impregnation of aqueous nickel salts followed by high-temperature sulfidation using $H_2S$, a flammable and highly toxic gas. Avoiding the excess use of hazardous reagents is another motivating factor in this research, in addition to reducing the environmental burden of catalyst production. Xanthogenate salts are prepared from $CS_2$ and the corresponding alcohol in stoichiometric equivalents, not requiring the large excess of reagent used in heterogeneous sulfidation reactions. This study will also demonstrate the advantage of an SSP approach to creating sulfided nickel catalysts over a method that simply replaces metallic platinum with metallic nickel.

Polyolefin materials represent a distinct challenge for heterogeneous catalysis due to their high molecular weights; mass and heat transfer are inefficient due to the high viscosity of most polymer melts, and long carbon chains hinder access to catalytic sites, notably the active site-containing micropores of zeolite catalysts that have proven effective in hydrocarbon transformations in industrial processes. To overcome some of these limitations, we have investigated various aluminosilicate supports, with varying degrees and types of porosity. Zeolite beta, molecular sieve 13X, MCM-41 and amorphous silica-alumina support were chosen for this initial survey as commercially available support materials with varying structural and acidic functionality.

One of the main challenges facing the hydrocracking of polymers is their physical properties, with molten viscous polymer creating mass and heat transfer limitations, which are then exacerbated by diffusion issues created by the use of a microporous zeolite [13,14]. Conventional catalysts, such as microporous zeolites, have proven to be effective catalysts in the hydrocracking of polymers, producing a significant fraction of saturated, branched paraffins [15–17]. Higher gas yields in the product stream can result because of hindered diffusion of large hydrocarbons through the narrow micropores of the catalyst [18]. Introducing larger mesopores (2–50 nm) enhances the diffusivity of molten polymer as well as heavy molecular weight molecules and increases the selectivity towards liquid product (for example, $C_5$-$C_{12}$) [5,19,20]. The preference for producing highly isomerized saturated naphtha is based on its high value in the petrochemical industry where it can be handled easily in comparison with gas, and it has direct application as a feedstock to a steam cracker unit, for the production of olefins and hydrogen.

Bifunctional catalysts are required in the hydrocracking process, consisting of zeolitic Brønsted acid sites [21] that facilitate the formation of carbonium ions [22], as well as metal sites that catalyse hydrogenation/dehydrogenation reactions [23]. Commonly, high surface area supports such as alumina, amorphous silica-alumina and zeolites are used to provide acidity and control turnover rates

[7–10]. The hydrogenation/dehydrogenation function can be provided by noble metals such as Pt or Pd or transition metals such as Mo, Co, Co-Mo, W, Ni or Ni-Mo [7]. Supported Pt catalysts have been found to exhibit high catalyst activity compared with the other metals by achieving high polymer degradation at comparatively lower temperatures and have been the subject of a significant recent study [24–27].

Catalysts often have a maximum content of 1 wt% Pt, but this small amount of noble metal still accounts for a significant proportion of the total operating cost [28,29]. Among the non-noble transition metals, Ni could be an effective component to functionalize the hydro-/dehydrogenating of different type of reactions [30,31]. However, the hydro-/dehydrogenation activity of Ni is lower than the Pt, so requires an increased Ni loading. A 10-fold increase in Ni still represents a reduced cost relative to platinum, since Pt (approx. \$40 000 kg$^{-1}$) is more than 2000 times more expensive than Ni (approx. \$18 kg$^{-1}$) in 2021 [29]. From a sustainability perspective, the earth abundance of Pt, Pd and other noble metals with catalytic applications limits their long-term mineral security, which may present problems in obtaining them for chemical processes. Beyond the field of catalysis, metal chalcogenides have found use in power storage and transmission as well as possessing unique optical and magnetic properties [32–34]. This provides options to move beyond noble metals in other technological applications, with the attendant benefits discussed above.

# 2. Material and methods

## 2.1. Materials

Impregnations were carried out using commercial zeolite beta (CP8134E Zeolyst International, Si/Al = 12.5), zeolite 13X (Sigma-Aldrich, Si/AL = 1.2), MCM-41 (Sigma-Aldrich) and amorphous silica-alumina catalyst support 135 (CAS no. 1335-30-4, Sigma-Aldrich Si/Al = 5.5). The generation of nickel sulfide and activation of the acidic aluminosilicate form was performed in a $H_2$ flow of 10 ml min$^{-1}$ at 480°C. Catalytic reactions of polymeric materials were performed at a polymer : catalyst ratio of 10 : 1, in a 300 ml stainless steel Parr reactor, with agitation by an 'anchor' style stirrer. Gas chromatography was performed using a Varian CP3800 fitted with 50 m × 0.32 mm i.d. PLOT $Al_2O_3$/ KCl capillary column (gaseous samples) and an Agilent Technologies 6890N Network GC fitted with a 100 m × 0.25 i.d. PONA CB column (liquid samples).

## 2.2. Precursor synthesis

Potassium O-ethylxanthate $K(S_2COC_2H_5)$ (3.53 g, 22 mmol) was dissolved in deionized water (120 ml). Nickel acetate tetrahydrate $Ni(O_2C_2H_3)_2 \cdot 4H_2O$ (2.49 g, 10 mmol) dissolved in deionized water (40 ml) was added dropwise and the green solution stirred for 3 h, forming a brown precipitate. After isolation by filtration under vacuum, the crude precipitate (4.56 g) was dissolved in chloroform (10 ml) and reprecipitated to yield flat dark green crystals of bis-O-ethylxanthate nickel (II) $Ni(S_2COC_2H_5)_2$ (1.95 g, 64.8%). Elem. Anal.: Found (calcd) for $C_6H_{10}N_2NiO_2S_4$: C, 24.1 (23.9); H, 3.2 (3.4); N, 19.4 (19.5). ESI-MS ($m/z$) 301 ([M + H], 100%), m.p. 136°C [35].

## 2.3. Catalyst characterization

The identity and crystallinity of the catalysts were determined using X-ray diffraction (Philips X' Pert Pro Diffractometer (PW3719) with Cu K$\alpha$ radiation ($\lambda$ = 1.54060 Å) generated by 10 mA current and 30 kV voltage. The $2\theta$ range was from 5°–60° using a scan speed of 0.06547° s$^{-1}$ and a step size of 0.0182521°. The Si/Al ratio as well as the added Ni and Pt level on the catalysts were determined using energy-dispersive X-ray fluorescence (EDXRF). The total surface area and micropore contribution of the catalysts was measured using BET (Quantachrome Quadrasorb EVO Kr/MP) at 77 K (electronic supplementary material, table S1). The samples were degassed at 350°C for 24 h. The acidity of the catalysts was quantified using ammonia temperature-programmed desorption (NH$_3$-TPD, Quantachrome ChemBET Pulsar TPR/TPD analyser). The modified zeolites were first activated/reduced under hydrogen flow of 40 cm$^3$ min$^{-1}$ at 480°C for 360 min. Following that, the catalysts were cooled to ambient temperature, then exposed to 5% of NH$_3$ in Ar and the temperature was ramped up to 800°C. Desorption of NH$_3$ is recorded against temperature and integrated to yield acidity profiles in terms of adsorption strength and correspond concentrations of different strength acid sites. Thermogravimetric analysis (TGA) was performed as follows: in flowing $N_2$ (40 ml min$^{-1}$), samples were heated at 5°C min$^{-1}$ to 600°C, pausing

**Table 1.** Metal concentrations and acidity properties of the catalysts used in this study. Metal concentrations were measured by EDXRF and SEM-EDX and Si/Al ratios were provided by the manufacturer and validated by EDXRF. Acid concentrations were measured be $NH_3$-TPD, as described in the experimental section.

| catalyst | metal conc. (actual wt%) (target wt%) | Si/Al Ratio | total acid concentration ($\mu mol_{NH3}.g^{-1}$) | weak acid concentration ($\mu mol_{NH3}.g^{-1}$) | strong acid concentration ($\mu mol_{NH3}.g^{-1}$) |
|---|---|---|---|---|---|
| 1Pt@Beta (WI) | 1.08 (1.0) | 12.5 | 1393 | 877 | 516 |
| 2.5Ni@Beta (WI) | 2.16 (2.5) | 12.5 | 1633 | 911 | 722 |
| 5Ni@Beta (WI) | 5.06 (5.0) | 12.5 | 1767 | 971 | 796 |
| 10Ni@Beta (WI) | 8.79 (10.0) | 12.5 | 1639 | 964 | 675 |
| 3Ni@SiAl catalyst support 135 (SSP) | 3.07 (1.0) | 5.5 | 533 | 413 | 120 |
| 5Ni@Beta (SSP) | 4.97 (2.0) | 12.5 | 722 | 563 | 159 |
| 2.5Ni@Beta (SSP) | 2.36 (1.0) | 12.5 | 759 | 582 | 176 |
| 8Ni@13X (SSP) | 8.04 (1.0) | 1.2 | 1385 | 1158 | 227 |
| 1Ni@MCM-41 (SSP) | 0.44 (1.0) | ∞[a] | 473 | 407 | 66 |

[a]Negligible alumina content.

for 60 min at 200°C and 400°C to equilibrate. At 600°C, the gas flow was switched to air (40 ml min$^{-1}$) and held at this temperature for 120 min to oxidize the remaining carbonaceous material. An example temperature programme and the corresponding mass losses associated with each temperature step is provided in the electronic supplementary material, Information.

## 2.4. Hydrocracking reaction

Hydrocracking of polymer was performed in a 300 ml stainless steel Parr reactor, agitated by an 'anchor' style stirrer, configured as described in a previous publication [11]. Polymer granules (10.00 g) and activated catalyst were combined in a weight ratio of 10 : 1 (polymer : catalyst). The reactor was sealed and purged three times with hydrogen before being charged to the reaction pressure (20 bar H$_2$). The reaction temperature of 330°C ± 5°C was reached within 45–55 min, and then it kept isothermally constant for 60 min with an agitation speed of 400 ± 25 r.p.m. After the reaction was completed, the reactor was cooled in flowing air to ambient temperature and the products were collected for analysis. The overall conversion was calculated using equation (2.1) where the conversion was considered as solid to fluid.

$$X_{(wt.\%)} = \frac{Solid_{out} - Polymer_{in} - Catalyst_{in}}{Polymer_{in}} \times 100. \tag{2.1}$$

Selectivity of products is calculated as a mass fraction of the conversion value, for given products of interest. The proportions of each product are evaluated separately in the gas phase and in the liquid phase, each collected after the reactor has been cooled to 20°C. These values are combined for volatile hydrocarbons present in both the liquid and gas phases at this temperature, i.e. C$_n$, where $5 > n > 10$.

The polymer feed comprised pure LDPE or a mixture of LDPE (34%), HDPE (24%), PP (32%) and PS (10%), obtained from Goodfellow Cambridge Ltd, with the exception of HDPE, derived from post-consumer packaging, specifically milk bottles.

## 3. Results

The conversion and selectivity of these catalysts is compared against the benchmark of previously published Pt@zeolite beta catalysts, tested at identical conditions, as well as Ni@zeolite synthesized via wet impregnation of Ni(NO$_3$)$_2$ . 6H$_2$O. In order to inform this comparison, table 1 provides a comparison of acidities, strongly dependent on both the support and the type of metal impregnated upon the support material. The amount of impregnated metal was also found to depend on both the

**Table 2.** Conversion of mixed plastic at 330℃, 60 min reaction time and an initial $H_2$ pressure of 20 bar, for catalysts used in this study. The feedstock used in these experiments is composed of LDPE (34%), HDPE (24%), PP (32%) and PS (10%), representing commercial mixed plastic waste.

| catalyst | $X_{Liquid}$ (wt%) | $X_{Gas}$ (wt%) | $X_{Solid\ to\ Fluid}$ (wt%) |
|---|---|---|---|
| 1Pt@Beta (WI) | 27.7 | 38.7 | 66.4 |
| 2.5Ni@Beta (WI) | 24.3 | 24.0 | 48.3 |
| 5Ni@Beta (WI) | 22.2 | 17.6 | 39.8 |
| 10Ni@Beta (WI) | 20.3 | 17.2 | 37.45 |
| 5Ni@Beta (SSP) | 28.9 | 64.5 | 93.4 |

support material and the precursor identity. Catalysts are referred to throughout according to their actual Ni loading, rather than the target loading.

Table 1 demonstrates the varying acidity properties of the acidic support materials, which are strongly correlated with the quantities of silica and alumina in the materials, as well as their overall porosity. A unit of $Al_2O_3$ can replace two units of $SiO_2$ in the framework and correspondingly create a Brønsted acid site. Zeolite beta, with a Si/Al ratio of 12.5, has a particularly high concentration of strong acid sites, and this property is also influenced by the quantity and identity of the metal that has been added to the support. The zeolitic crystal structure is partially disordered but provides a high internal specific surface area, relative to amorphous aluminosilicates and mesoporous silicates such as MCM-41.

Strongly acidic sites are important for stabilizing carbocation intermediates in zeolites [36], and a higher proportion of alumina units in the zeolite framework increase the concentration of these sites [37]. Beyond a Si/Al ratio of 15, a sufficient concentration of strongly acidic sites is evident for the turnover rates achieved in this study, although even completely siliceous supports maintain enough acidity for facilitating polyolefin hydrocracking. Metal impregnation has a significant effect on the overall acidity of the catalyst, and the method of impregnation is also significant. Wet impregnation with $Ni(NO_3)_2$ increases both strong acid and weak acid concentrations. This may be due to incomplete removal of nitrate ions from the framework during catalyst activation by calcination, or the nitrate assisting in the ion exchange of residual Na in the original zeolite structure by salt metathesis to produce Brønsted acid sites. By contrast, SSP impregnation of the supports can attenuate acidity, and doubling the amount of Ni causes a 5% drop in total acidity in the case of the zeolite beta support. When considering this and the surface area measurements presented in the electronic supplementary material, table S1, it is clear that there is some loss of micropore volume, and therefore fewer acid sites situated within micropores are observed at higher loadings of Ni SSP. This could become a limiting factor in scale-up and may mean that a lower Ni level is selected with different operating conditions, rather than high Ni concentrations and milder conditions.

The conversion and selectivity of each of these catalysts are compared against the benchmark of previously published Pt@zeolite beta catalysts, tested at identical conditions, as well as Ni@zeolite synthesized via wet impregnation of $Ni(NO_3)_2 \cdot 6H_2O$. Table 2 demonstrates that the overall conversion, as well as the proportions of liquid and gas obtained, from identical experiments can vary significantly by changing the catalyst support and the quantity of metal loaded onto the catalyst support. While a high level of acidity is an important factor in stabilizing carbocations in the later stages of this reaction, it is clear by comparison of tables 1 and 2 that the identity and concentration of the metal is key to obtaining high levels of conversion. This is due to the hydrogenation/dehydrogenation function provided by the metal that is the dominant process in molecular weight reduction for bulky hydrocarbons such as polymers. Only after a sufficient degree of conversion can the molecules enter the mesopores of a heterogeneous catalysis, and an even greater degree of conversion is required for species to enter the micropores of zeolite-type materials. We therefore conclude that the most important parameter for catalyst optimization is the identity and concentration of the metal species.

Considering the selectivity to liquids and gases, as well as the overall conversion of the starting polymer, is important when selecting a catalyst for this hydrocracking process. As previous studies have indicated, 1Pt@Beta is a highly selective catalyst for the production of butane, pentane and hexane, preferentially generating branched isomers of these saturated hydrocarbons in the

**Table 3.** Conversion of LDPE at 330°C, 60 min reaction time and an initial $H_2$ pressure of 20 bar, for catalysts used in this study, in order to screen the catalyst activity prior to testing the catalyst in the conversion of mixed polyolefin streams.

| catalyst | $X_{Liquid}$ (wt%) | $X_{Gas}$ (wt%) | $X_{Solid\ to\ Fluid}$ (wt%) |
| --- | --- | --- | --- |
| 3Ni@SiAl catalyst support 135 (SSP) | 74.3 | 22.8 | 97.1 |
| 5Ni@Beta (SSP) | 38.0 | 57.2 | 95.2 |
| 2.5Ni@Beta (SSP) | 0 | 28.5 | 28.5 |
| 8Ni@13X (SSP) | 0 | 18.6 | 18.6 |
| 1Ni@MCM-41 (SSP) | 0 | 21.5 | 21.5 |

hydrocracking of LDPE. Replicating the behaviour of this system with earth-abundant transition metals is the aim of this study, as this fraction (commonly referred to as naphtha) is a valuable feedstock that is otherwise commonly produced by the Fischer–Tropsch process, an energy-intensive method of converting CO and $H_2$ (or $H_2O$) into higher hydrocarbons. Hydrocracking polymer feedstocks and limiting the decomposition of carbon chains is an alternative to gasification of municipal solid waste, only to then convert it into higher hydrocarbons again with the complications involved [38].

Other aluminosilicates, seen in table 3, were screened to replicate this behaviour but offered poor selectivity in the case of SiAl catalyst support 135, or poor conversion in the case of MCM-41 and zeolite 13X. Zeolite beta was therefore the support material of choice for further investigations in this study. Note that screening was performed with pure LDPE, to remove the complications of mixed material feeds in identifying catalysts with high hydrocracking activity. The key catalyst in this study 5Ni@Beta (prepared by a SSP route) was tested with both pure LDPE and a mixed polyolefin feedstock to demonstrate its effectiveness in both experimental conditions. Beyond the liquid and gas selectivity, the precise product selectivity was changed significantly by the feedstock, particularly the presence of PS, and its aromatic components that are resistant to hydrogenation in these conditions.

As previously noted, zeolite beta supports exhibit a marked selectivity toward $C_4$ hydrocarbons in the hydrocracking of polyolefins. The slightly higher conversion towards gases in the case of 2.5 wt% SSP-impregnated zeolite beta, 2.5Ni@Beta, is (28.5%) relative to 5 wt% SSP-impregnated zeolite beta, 5Ni@Beta (22.8%) accounts for the higher proportions of all gas products shown in figure 1; however, the selectivity pattern was identical when normalized against the total amount of gas in the sample. Conversely, the silica-alumina catalyst support 135 produces undesirable methane and ethane fractions from this reaction, although it provides high conversion overall. In line with the high acidity of this catalyst support (table 1), almost full conversion (97.1%) was achieved, but without a highly ordered crystalline structure, the product distribution is not as focused as the case of the zeolite catalyst supports. As no liquid products were isolated from 2.5Ni@Beta at these conditions, no direct comparison was possible. Figure 2 compares the available liquid product distributions for the catalysts prepared by SSP impregnation, i.e. 3Ni@CS135 and 5Ni@Beta with LDPE and with mixed polyolefins.

The liquid product distributions were determined by gas chromatography–mass spectrometry (GC-MS) (figure 2) and demonstrate the selectivity of the beta support towards saturated hydrocarbons when converting LDPE into gas and liquid products. Table 3 provides this data numerically. The selectivity towards a given hydrocarbon is highly dependent on both the support and the product stream, as mixed plastics tend to contain additional function groups. The pore size of zeolite beta favours the formation of $C_4$-$C_6$ hydrocarbons, with butane and its isomers being present in the gas products of this reaction, collected at 25°C and ambient pressure. Further discussion of the distribution of butane isomers is provided in the Discussion section, table 4.

## 4. Discussion

Sulfidation of catalysts has to be tailored to the catalytic metal of interest [39] and is known to affect the activity and stability of catalyst materials and their supports [40]. Removing this step from processing is desirable as it also removes the need for operational facilities to use hazardous sulfiding agents, such as hydrogen sulfide, dimethyl sulfide or tertiary-butyl polysulfide. The selected precursor $Ni[S_2OCH_2CH_3]_2$ does not form stoichiometric $Ni_xS_y$ compounds on zeolite beta at the activation conditions used in this

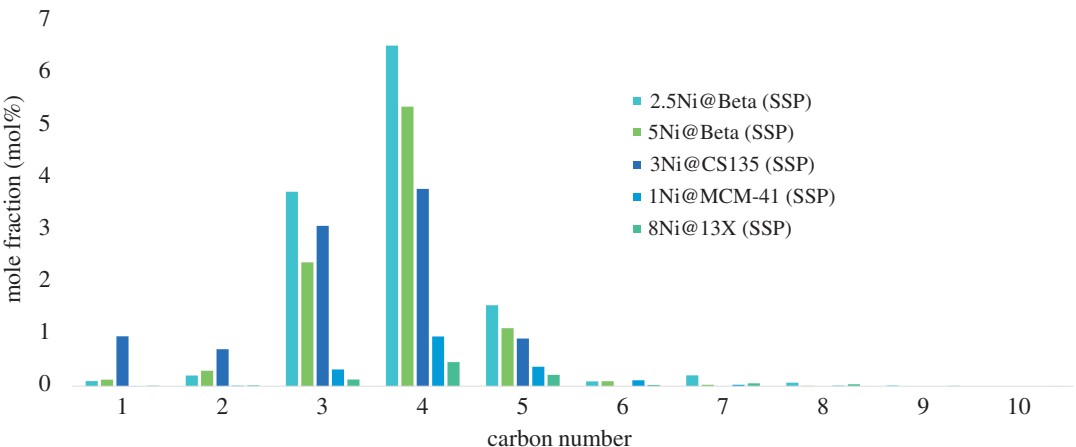

**Figure 1.** Selectivity of SSP-derived Ni@Aluminosilicate catalysts in mol% of the gaseous components collected after reaction of LDPE at 330℃ with an initial H₂ pressure of 20 bar. The remainder of the gas sample is H₂, and liquid analysis was performed separately.

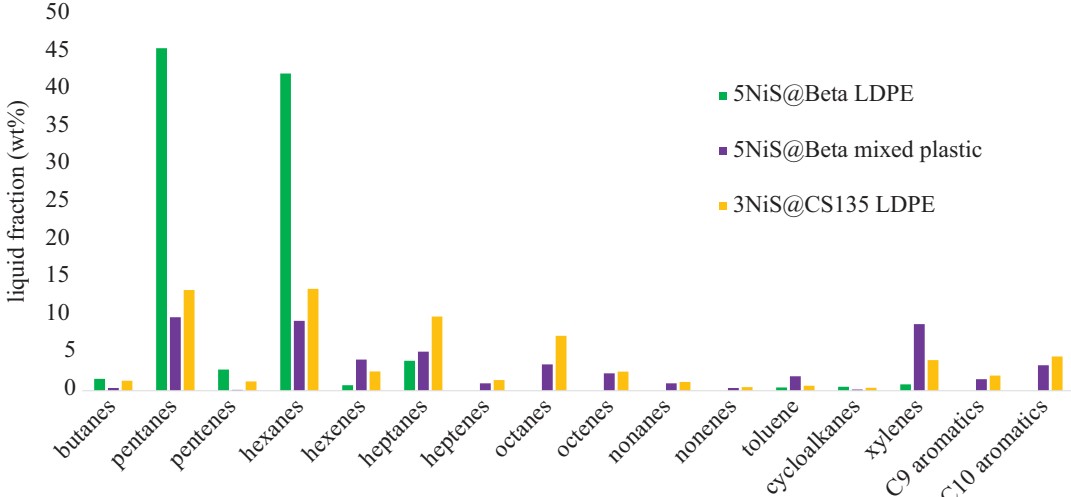

**Figure 2.** Selectivity of liquid hydrocarbon products determined by GC-MS. Products were collected as liquid samples after reaction of LDPE at 330℃ with an initial H₂ pressure of 20 bar. Compound groups were identified by mass spectrometry using the molecular ion and splitting pattern of daughter ions.

study, due to the high temperatures and presence of hydrogen, but the low concentrations of sulfur provide a significant improvement in conversion relative to an Ni@beta catalyst prepared via wet impregnation and subsequent reduction in hydrogen. Sulfur vacancies are known to have a significant effect in transition metal sulfided catalysts [41], and these structural defects are active sites for catalytic reactions. Supported nickel sulfide is sensitive to preparation conditions, tending to form preferentially on the external surface of zeolite supports, and it has been demonstrated that the catalytic activity of Ni@Y for hydrodesulfurization reactions varies significantly depending upon whether wet impregnation or ion exchange is used to prepare the catalyst [42]. Wet impregnation offers a superior coking resistance to ion exchange, and greater selectivity to C₄ hydrocarbons in hydrodesulfurization of thiophene, attributed to high dispersion of the metal on the support. This study demonstrates that selectivity can also be dominated by the support material when considering the conversion of polymeric materials, in addition to the method of impregnation.

The levels of Ni impregnation given in table 5 illustrate the difficulty of achieving the target concentration of metal between different precursors. 2.5Ni@Beta (SSP) and 5Ni@Beta (SSP) were found to have 2.36 and 4.97 wt% of Ni respectively, measured by SEM-EDX. While not a strictly surface analysis technique, the limited penetration of accelerated electrons provided by the SEM beam provide a bias towards surface analysis in the EDX analysis of emitted X-rays. For this reason, the level of surface species will be over-represented versus the bulk composition. The precursor Ni(S₂COC₂H₅)₂ is

**Table 4.** Liquid product selectivity to hydrocarbon components, determined by GC-MS. Selectivity is provided as a wt% of the overall liquid sample, collected at 25℃ and ambient pressure from the reactor.

| compound | 5NiS@Beta LDPE | 5NiS@Beta mixed plastic | 3NiS@CS135 LDPE |
| --- | --- | --- | --- |
| butanes | 1.6 | 0.4 | 1.3 |
| pentanes | 45.6 | 9.8 | 13.4 |
| pentenes | 2.8 | 0.1 | 1.2 |
| hexanes | 42.2 | 9.3 | 13.6 |
| hexenes | 0.7 | 4.2 | 2.6 |
| heptanes | 4.0 | 5.2 | 9.9 |
| heptenes | — | 1.0 | 1.4 |
| octanes | — | 3.5 | 7.3 |
| octenes | — | 2.3 | 2.5 |
| nonanes | — | 1.0 | 1.2 |
| nonenes | — | 0.4 | 0.5 |
| toluene | 0.4 | 1.9 | 0.7 |
| cycloalkanes | 0.5 | 0.2 | 0.4 |
| xylenes | 0.9 | 8.9 | 4.1 |
| $C_9$ aromatics | — | 1.5 | 2.0 |

**Table 5.** Metal concentrations of the catalysts used in this study. Metal concentrations were measured by SEM-EDX.

| catalyst | Ni (wt%) | S (wt%) | Ni (at%) | S (at%) | Ni : S atomic ratio |
| --- | --- | --- | --- | --- | --- |
| 2.5Ni@Beta (SSP) | 2.36 | 0.12 | 4.02 | 0.37 | 10.76 |
| 5Ni@Beta (SSP) | 4.97 | 0.21 | 8.47 | 0.65 | 12.94 |
| 3Ni@SiAl catalyst support 135 (SSP) | 3.07 | 0.10 | 5.23 | 0.31 | 16.79 |
| 8Ni@13X (SSP) | 8.04 | 0.15 | 13.70 | 0.47 | 29.32 |
| 1Ni@MCM-41 (SSP) | 0.44 | 0.25 | 0.75 | 0.78 | 0.96 |

significantly larger than the simple ionic compound $Ni(NO_3)_2$ and is therefore less likely to enter the pores of the zeolite substrate, leading to preferential deposition of metal on the exterior surfaces of catalyst particles.

The substoichiometric nickel sulfides presented herein could not be identified by pXRD due to their low concentration in the catalyst relative to the support; however, elemental analysis provided by SEM-EDX reveals that the Ni : S concentration is typically between 10 and 20, corresponding to a stoichiometry of $NiS_{0.1>x>0.05}$. There are a wide variety of nickel sulfide phases; however, this ratio of Ni : S is higher than commonly reported stoichiometric crystalline phases. $Ni_3S_2$, $Ni_3S_4$, $Ni_7S_6$, $Ni_9S_8$, $NiS_2$, $NiS_{1.03}$ and NiS have all been reported as catalytically active metastable phases, with $NiS_{1.03}$ performing particularly well in cyclohexanone hydrogenation [43]. The compounds formed in this study are therefore likely to be disordered materials with sulfur distributed randomly throughout the nickel bulk and surface, further reducing the likelihood of their detection by bulk diffraction techniques. Further experimental work would be required to determine the distribution of oxidation states in this material, as well as the nature of sulfur inclusion in its structure, but it is clear that the inclusion of sulfur vastly enhances its catalytic activity relative to catalysts prepared by wet impregnation of nickel on zeolite beta.

The precursor used in this study is known to produce stoichiometric $Ni_xS_y$ compounds by thermolysis; however, the presence of hydrogen in the activation of this catalyst creates a reductive environment and consequently alters the thermolysis behaviour of this compound. Lower oxidation states of nickel are produced by reductive hydrogen, producing metallic Ni(0) from Ni(II) found in

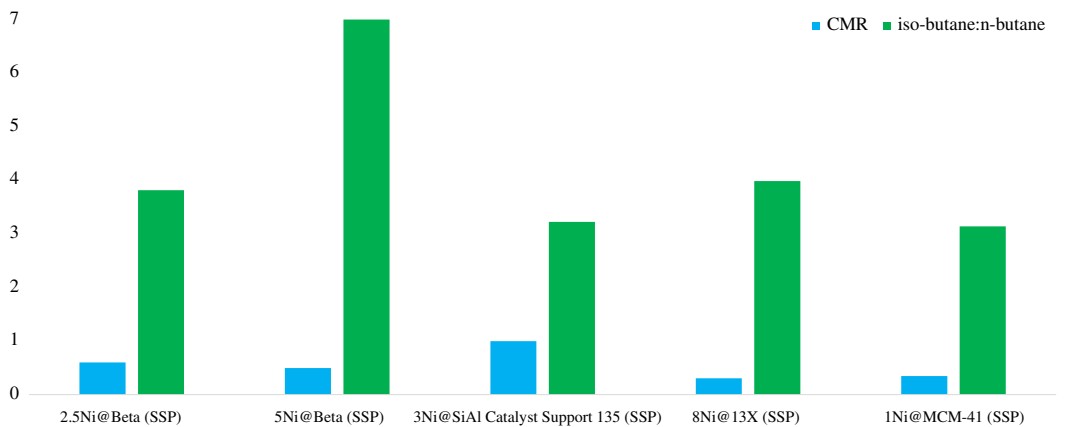

**Figure 3.** CMR and iso-butane : n-butane ratio for SSP-derived catalysts. Products are from the reaction of LDPE at 330°C for 60 min with an initial $H_2$ pressure of 20 bar.

the bis-O-ethylxanthate nickel (II) complex. Specific ligand systems are required to access higher oxidation states such as Ni(III) and Ni(IV) [44], so these states are not considered herein. Different catalyst supports were found to affect the reducibility of Ni, as evidenced by the varying Ni:S ratio. This may be due to the strength with which the precursor adsorbs to the surface during impregnation, as well as the ability of Si-O$^-$ and Al-O$^-$ species surface groups to stabilize Ni(II).

To provide further insight into the cracking mechanisms and consequent selectivity of the various SSP-derived catalysts, cracking mechanism ratio (CMR) and iso-butane: n-butane ratios were calculated from the analysis of products formed during hydrocracking. Figure 3 illustrates the significant variation of these quantities, dependent upon the catalyst support. CMR is defined by equation (4.1) and indicates whether the reaction is dominated by protolytic cracking or β-scission, facilitated by acid sites within the zeolite support [45]. When CMR > 1, this indicates that protolytic cracking is dominant in the final product selectivity. In the case of a bifunctional catalyst, the accessibility of metal sites for the fission of hydrogen will increase the rate of monomolecular protolytic cracking.

$$CMR = \frac{(C_1 + \sum C_2)}{iC_4}. \tag{4.1}$$

Isomerization can occur within zeolite pores and at active metal sites, and it should also be noted that the LDPE feedstock contains tertiary carbon environments prior to depolymerization. Figure 3 uses the ratio of iso-butane : n-butane to provide an indication of whether isomerization occurs for all hydrocarbon compounds and demonstrates that while it is prevalent in all reactions using the catalysts discussed above, it is particularly notable in the case of 5Ni@Beta (SSP) due to its increased metal loading relative to the other catalysts. Iso-butane is frequently used in petrochemical alkylation reactions [46,47] and is therefore considered a valuable fraction of refinery intermediates.

Polyolefins are defined by their C-C bonded polymeric backbone (figure 4) but can contain a wide variety of substituents. The most relevant of these in terms of residential and commercial waste streams are polyethylene, PP and PS, as they are used extensively in packaging, accounting for 29.7%, 19.3% and 6.4%, respectively, in Europe of total plastic waste including diverse polymers beyond polyolefins [48,49]. PS was included in this study as it can account for around 10% of the polyolefin content of plastic waste streams in the UK, and its aromatic substituents provide the possibility of diverse chemistry in the hydrocracking product stream and has received interest as a feedstock for other chemical reclamation processes [50,51]. Aromatic compounds are observed at approximately the level of PS addition into the feed, accounting for 15.7 wt% of the liquid products produced in the hydrocracking of the mixed feed detailed herein catalysed by 5Ni@Beta (SSP), corresponding to 11.9% of the overall gas and liquid product mixture in this reaction. This catalyst produces a fraction of aromatic compounds (1.3 wt%) from pure LDPE, indicating that reforming reactions are occurring within the catalyst, and 3Ni@CS135 produces a significant aromatic fraction of 11.3 wt% from pure LDPE under the reaction conditions used in this study.

Separation of aromatic-containing hydrocarbons such as benzene, toluene, xylenes and higher alkylbenzenes can be performed by distillation, but it is advantageous for material to be sorted prior

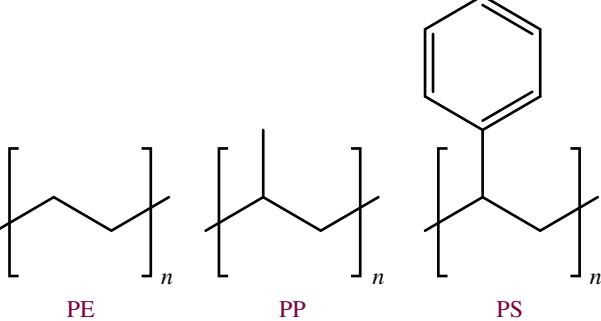

**Figure 4.** Polymer repeat units of the polyolefins studied as feedstocks for polymer recycling.

to hydrocracking to minimize the need for this kind of post-treatment. Previous work on spent FCC catalysts of low acidity, and contaminated with heavy metals [17,52], has demonstrated that in the absence of hydrogen, the product stream contains an excess (up to 80%) of unsaturated hydrocarbons. Scale-up of this process would therefore consider catalyst selection as well as the benefits of high hydrogen pressures of greater than 20 bar and its associated operational cost, against the advantages of a paraffinic product selection. Equally, oxidative processes such as steam-cracking can be used to provide C=C functionality, once depolymerization has been performed by hydrocracking to take advantage of the high $C_4$–$C_6$ selectivity of the zeolite beta support.

PS is an ideal feedstock to create these aromatic compounds, which can act as a platform chemical for other substituted aromatics in a wider economic model that considers waste plastic. The catalysts presented herein offer an alternative to extensive material sorting, as PS is not only tolerated by the catalyst, but produces a rich product stream. This creates a resource not just for recycling, but as a substitute to fossil-based feedstocks. Recovery of vinyl benzene (aka styrene) for PS production would allow total circularity of polymer production and recovery, and has been attempted by many methodologies including pyrolysis [53], supercritical solvothermolysis [54] and supercritical water partial oxidization [55]. Equally, the aromatic products produced in catalytic hydrocracking have other applications, such as xylenes used to synthesize polybenzimidazoles [56], PET [57] and metal-organic frameworks [58].

The degree to which a catalyst forms insoluble coke on its surface is highly dependent upon its resistance to the formation of large aromatic molecules [59], known as coke precursors [60]. This process is a common deactivation mechanism for bifunctional catalysts in all hydrocarbon transformations [60–62] and is mitigated by higher hydrogen pressures, and ultimately regeneration of the catalyst. Figure 4 demonstrates the dependence, in these experiments, of coke level at the end of a single reaction upon the level of conversion obtained in the reaction itself. Pt catalysts are notably resistant to coke formation, as seen by the high coke : conversion ratio of 1Pt@Beta (WI) in figure 5, and this is one of the many properties of noble metals that have made them difficult to replace in these catalytic applications. The values presented in figure 5 determined by TGA correspond to the weight percentage loss of the catalyst at 600°C in the presence of flowing air. Prior to this, heating is performed in stages under an inert $N_2$ flow, which removes residual volatiles, adsorbed water and pyrolyses the polymer associated with the catalyst, leaving only coke precursors and insoluble coke. Switching to an oxygen-containing gas at this stage facilitates the combustion of coke, and the final catalyst weight can be compared against the weight loss in this final stage. Tables of values and a sample TGA trace are provided in the electronic supplementary material, Information.

The resistance to coking is not only a property of the catalyst, but of the process, and this study recognizes that optimization of catalysed polymer hydrocracking relies not only on innovations in catalyst technology, but in driving the entire process to greater economic viability. The reactions studied here are in batch conditions, whereas other process configurations such as fluidized beds may provide greater heat- and mass-transfer possibilities that might maximize catalytic turnover and reduce over-reaction of polymer molecules that can generate coke precursors. An increased hydrogen pressure would also drive production of volatile molecules that are less likely to become entrained as coke. The high conversions demonstrated by 2.5Ni@Beta (SSP), 5Ni@Beta (SSP) and 3Ni@SiAl catalyst support 135 (SSP) could also lead to the formation of coke precursors after completion of depolymerization reactions, as shown in figure 5, so coke formation could be minimized by continuous processing of polymer and extraction of volatile molecules as they form. Hydrocracking of

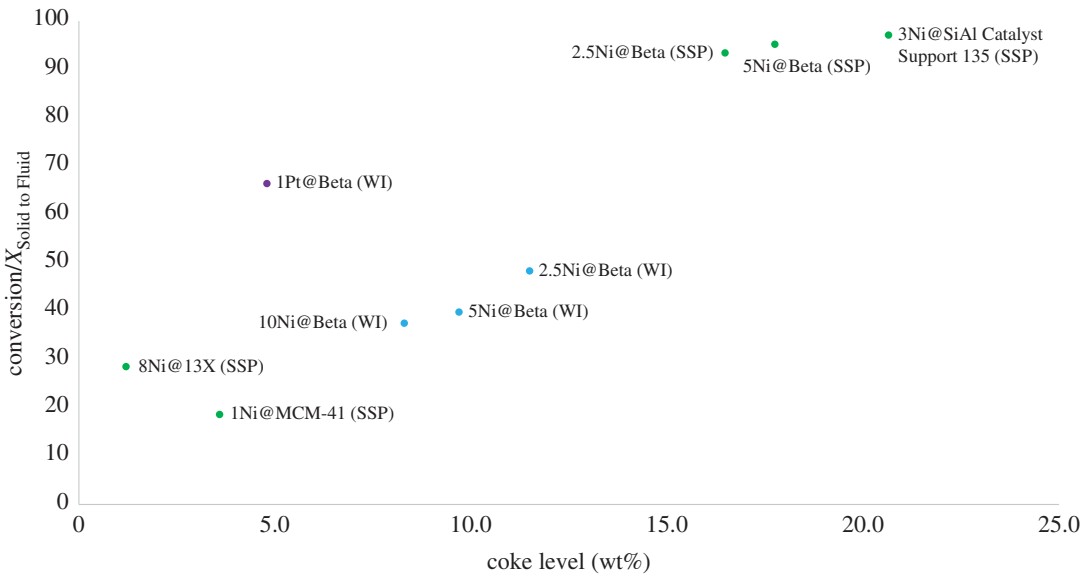

**Figure 5.** Coke level determined by TGA of the catalysts used in this study post-reaction.

heavy oils typically use process configurations of fixed or ebullated bed reactors, a type of fluidized bed reactor. The ebullated bed reactors allow for more uniform distribution of coke through the catalyst inventory providing more uniform deactivation. In addition, catalyst is added and removed continuously in order to maintain catalytic activity at a certain constant level. The high viscosity of polymer melts, even significantly above their melting temperature, must be considered in reactor design, and the agitation method chosen in this study maximizes polymer–catalyst contact in batch conditions. Continuous processing presents additional challenges, but also opportunities for optimization when considering problems such as coke formation.

# 5. Conclusion

The hydrocracking of polyolefin waste is a method of chemical recycling that conserves carbon–carbon bonds, and appropriate catalyst selection allows tailoring of the product stream despite vastly different compositions of plastic feedstocks. The catalyst presented herein builds upon the success of platinum zeolite catalysts in this application and demonstrates that while nickel is a suitable replacement, it must be appropriately deposited on the zeolite support and sulfided to provide the high levels of activity required for this process. To this end, a robust SSP approach is presented, capable of delivering nickel and sulfur simultaneously via solvent impregnation followed by thermolysis.

The selectivity toward gas and liquid products, as well as the identities of components within these fractions and their respective selectivities are presented here. The benefit of microporous aluminosilicates i.e. zeolites is an enhanced selectivity to light hydrocarbons in the naphtha range i.e. $C_4$–$C_6$, and it is demonstrated that an increase of Ni loading from 1 wt% on the support to 2 wt% can dramatically improve conversion (greater than 95 wt%) at the conditions presented, while preserving selectivity. 5Ni@Beta(SSP) generates saturated $C_4$ (37.3 wt%), $C_5$ (21.6 wt%) and $C_6$ (12.8 wt%) hydrocarbons from a feed of pure LDPE, and these fractions are collected in both gas and liquid phases.

It is anticipated that further research into the optimized operating conditions of this catalyst can minimize the formation of coke precursors and coke in the micropores of the zeolite and on the surface of both the support and the supported metal. The established use of aluminosilicate-supported nickel sulfides for hydrodesulfurization [63,64] and hydrodenitrogenation [65] makes this a promising catalyst for tolerating sulfur and nitrogen heteroatom poisons that may be present in polymer waste streams. Due to the diversity of polymers, additives and other types of waste (e.g. food, agricultural) that could contaminate even relatively pure polyolefin waste streams, this is an important consideration for advancing the technology readiness level of any chemical recycling technology for polymers beyond uncatalysed pyrolysis.

Ethics. This research did not involve experiments performed on animals or humans.

Data accessibility. The datasets supporting this article have been uploaded as part of the electronic supplementary material [66].

Authors' contributions. A.A.T.: conceptualization, data curation, formal analysis, investigation, methodology, project administration, writing—original draft and writing—review and editing; A.B.J.: investigation, methodology and writing—review and editing; E.A.: formal analysis, investigation, methodology and writing—review and editing; A.A.G.: conceptualization, formal analysis, funding acquisition, investigation, project administration, resources, supervision and writing—review and editing.

All authors gave final approval for publication and agreed to be held accountable for the work performed therein.

Competing interests. We declare we have no competing interests.

Funding. UK Catalysis Hub is kindly thanked for resources and support for Dr Tedstone via our membership of the UK Catalysis Hub Consortium and funding through EPSRC grant no. EP/R027129/1. The authors would also like to acknowledge the support of King Saud University in Riyadh (KSU) for research funding.

Acknowledgements. The authors would like to thank the contribution of Dr Desmond Doocey, Dr Cameron Price and Dr Sarayute Chansai in collecting experimental data for this study.

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
