## [Peer Review File · Royal Society Open Science]

Review History

RSOS-211353.R0 (Original submission)

Review form: Reviewer 1

Is the manuscript scientifically sound in its present form?

Yes

Are the interpretations and conclusions justified by the results?

Yes

Is the language acceptable?

Yes

Do you have any ethical concerns with this paper?

No

Have you any concerns about statistical analyses in this paper?

No

Recommendation?

Major revision is needed (please make suggestions in comments)

Comments to the Author(s)

The author synthesized nickel based catalyst deposited on the zeolite support followed with sulfide treatment. As prepared catalyst exhibits excellent selectivity toward gas and liquid products and has certain application prospects, it's recommended to publish in the Royal Society Open Science.

(1) The sample should be defined when it first appeared.

(2) Why does the as-prepared catalyst have excellent selectivity? More characterizations and discussion should be provided.

(3) The performance of the Pt@Beta (WI) catalyst with different proportions should be investigated.

(4) Metal chalcogenides are promising materials for the next generation of electronic devices. It not only can be used as catalysts for heterogeneous catalytic reactions, but also can be used as electrode materials for batteries and supercapacitors. I suggest authors cite or highlight some recent important progresses in this field.

Review form: Reviewer 2

Is the manuscript scientifically sound in its present form?

Yes

Are the interpretations and conclusions justified by the results?

Yes

Is the language acceptable?

Yes

Do you have any ethical concerns with this paper?

No

Have you any concerns about statistical analyses in this paper?

No

Recommendation?

Major revision is needed (please make suggestions in comments)

Comments to the Author(s)

The present manuscript describes the synthesis and characterization of beta-supported nickel (Ni@Beta) hydrocracking and hydrotreating catalyst, as an alternative to wet impregnation using aqueous nickel (II) nitrate, to provide catalytic materials with higher conversion to fluid products in hydrocarbon refining, at the same mild batch reaction conditions of 330 °C with appropriate agitation and 20 bar H₂ pressure. The authors demonstrated that the 5wt%Ni@Beta yielded a >95 wt% conversion of a mixed polyolefin feed to fluid products even though there were coke formation issues that could be investigated further in the future. The catalysts have been characterised by, PXRD, EDX, BET, TGA and NH₃-TPD. The products have been identified by GC-MS. The preparation of the compounds is clear while the manuscript is in general well written and referenced.

The comparison of acidities hasn't been discussed very clearly. Even though the concentration of metal is a crucial factor how the acidities contribute to the performance of the catalytic process? What information we can extract from table 1? Also if would worth including the parameters in order of higher impact on the specific catalytic process. Porosity? Transition metal? Operating temperature? Acidity? The authors might want to improve this part of the discussion.

The pXRD in the supplementary information (figure 2) reads "pXRD of selected catalysts demonstrating the preservation of crystal structures." It is not clear what the authors indicate here by "preservation". Is it before and after the catalytic cycle? I might have missed something here but, if the authors want to show any kind of preservation, they need to compare it with something else.

The authors mentioned that the catalysts have been characterised by BET but there are no data in the manuscript or SI. What surface areas have been identified? How do they compare with other catalysts?

This is an interesting piece of work with the main highlight in my opinion is the demonstration of the importance of preparation method of the catalyst which led to substantially higher performance without the use of noble metals even though some coking issues are present. I do believe that the manuscript can be potentially appropriate for publication in Royal Society open science but only after major revision and clarification of the points raised above.

Decision letter (RSOS-211353.R0)

Dear Dr Tedstone:

Title: Transition Metal Chalcogenide Bifunctional Catalysts for Chemical Recycling by Plastic Hydrocracking: A Single Source Precursor Approach
Manuscript ID: RSOS-211353

The editor assigned to your manuscript has now received comments from reviewers. We would like you to revise your paper in accordance with the referee and Subject Editor suggestions which can be found below (not including confidential reports to the Editor). Please note this decision does not guarantee eventual acceptance.

Please submit your revised paper before 22-Oct-2021. Please note that the revision deadline will expire at 00.00am on this date. If we do not hear from you within this time then it will be assumed that the paper has been withdrawn. In exceptional circumstances, extensions may be possible if agreed with the Editorial Office in advance. We do not allow multiple rounds of revision so we urge you to make every effort to fully address all of the comments at this stage. If deemed necessary by the Editors, your manuscript will be sent back to one or more of the original reviewers for assessment. If the original reviewers are not available we may invite new reviewers.

To revise your manuscript, log into <http://mc.manuscriptcentral.com/rsos> and enter your Author Centre, where you will find your manuscript title listed under "Manuscripts with

Decisions." Under "Actions," click on "Create a Revision." Your manuscript number has been appended to denote a revision. Revise your manuscript and upload a new version through your Author Centre.

Yours sincerely,
Dr Ellis Wilde
Publishing Editor, Journals

RSC Associate Editor
Comments to the Author:
(There are no comments.)

RSC Subject Editor
Comments to the Author:
(There are no comments.)

Reviewers' Comments to Author:

Reviewer: 1

Comments to the Author(s)

The author synthesized nickel based catalyst deposited on the zeolite support followed with sulfide treatment. As prepared catalyst exhibits excellent selectivity toward gas and liquid products and has certain application prospects, it's recommended to publish in the Royal Society Open Science.

(1) The sample should be defined when it first appeared.

(2) Why does the as-prepared catalyst have excellent selectivity? More characterizations and discussion should be provided.

(3) The performance of the Pt@Beta (WI) catalyst with different proportions should be investigated.

(4) Metal chalcogenides are promising materials for the next generation of electronic devices. It not only can be used as catalysts for heterogeneous catalytic reactions, but also can be used as

electrode materials for batteries and supercapacitors. I suggest authors cite or highlight some recent important progresses in this field.

Reviewer: 2

Comments to the Author(s)

The present manuscript describes the synthesis and characterization of beta-supported nickel (Ni@Beta) hydrocracking and hydrotreating catalyst, as an alternative to wet impregnation using aqueous nickel (II) nitrate, to provide catalytic materials with higher conversion to fluid products in hydrocarbon refining, at the same mild batch reaction conditions of 330 °C with appropriate agitation and 20 bar H₂ pressure. The authors demonstrated that the 5wt%Ni@Beta yielded a >95 wt% conversion of a mixed polyolefin feed to fluid products even though there were coke formation issues that could be investigated further in the future. The catalysts have been characterised by, PXRD, EDX, BET, TGA and NH₃-TPD. The products have been identified by GC-MS. The preparation of the compounds is clear while the manuscript is in general well written and referenced.

The comparison of acidities hasn't been discussed very clearly. Even though the concentration of metal is a crucial factor how the acidities contribute to the performance of the catalytic process? What information we can extract from table 1? Also if would worth including the parameters in order of higher impact on the specific catalytic process. Porosity? Transition metal? Operating temperature? Acidity? The authors might want to improve this part of the discussion.

The pXRD in the supplementary information (figure 2) reads "pXRD of selected catalysts demonstrating the preservation of crystal structures." It is not clear what the authors indicate here by "preservation". Is it before and after the catalytic cycle? I might have missed something here but, if the authors want to show any kind of preservation, they need to compare it with something else.

The authors mentioned that the catalysts have been characterised by BET but there are no data in the manuscript or SI. What surface areas have been identified? How do they compare with other catalysts?

This is an interesting piece of work with the main highlight in my opinion is the demonstration of the importance of preparation method of the catalyst which led to substantially higher performance without the use of noble metals even though some coking issues are present. I do believe that the manuscript can be potentially appropriate for publication in Royal Society open science but only after major revision and clarification of the points raised above.

Author's Response to Decision Letter for (RSOS-211353.R0)

See Appendix A.

RSOS-211353.R1 (Revision)

Review form: Reviewer 2

Is the manuscript scientifically sound in its present form?

Yes

Are the interpretations and conclusions justified by the results?

Yes

Is the language acceptable?

Yes

Do you have any ethical concerns with this paper?

No

Have you any concerns about statistical analyses in this paper?

No

Recommendation?

Accept with minor revision (please list in comments)

Comments to the Author(s)

The authors made an effort to address adequately the reviewers' concerns. Minor point: Please format the reference list according to the journal's formatting guidelines. I am happy to recommend publication after minor revision.

Decision letter (RSOS-211353.R1)

Dear Dr Tedstone:

Title: Transition Metal Chalcogenide Bifunctional Catalysts for Chemical Recycling by Plastic Hydrocracking: A Single Source Precursor Approach
Manuscript ID: RSOS-211353.R1

Thank you for submitting the above manuscript to Royal Society Open Science. On behalf of the Editors and the Royal Society of Chemistry, I am pleased to inform you that your manuscript will be accepted for publication in Royal Society Open Science subject to minor revision in accordance with the referee suggestions. Please find the reviewers' comments at the end of this email.

The reviewers and handling editors have recommended publication, but also suggest some minor revisions to your manuscript. Therefore, I invite you to respond to the comments and revise your manuscript.

Please also include the following statements alongside the other end statements. As we cannot publish your manuscript without these end statements included, if you feel that a given heading is not relevant to your paper, please nevertheless include the heading and explicitly state that it is not relevant to your work. We have included a screenshot example of the end statements for reference.

- Ethics statement

Please clarify whether you received ethical approval from a local ethics committee to carry out your study. If so please include details of this, including the name of the committee that gave

consent in a Research Ethics section after your main text. Please also clarify whether you received informed consent for the participants to participate in the study and state this in your Research Ethics section.

OR

Please clarify whether you obtained the necessary licences and approvals from your institutional animal ethics committee before conducting your research. Please provide details of these licences and approvals in an Animal Ethics section after your main text.

OR

Please clarify whether you obtained the appropriate permissions and licences to conduct the fieldwork detailed in your study. Please provide details of these in your methods section.

- Data accessibility

It is a condition of publication that you make available the data and research materials supporting the results in the article. Datasets should be deposited in an appropriate publicly available repository and details of the associated accession number, link or DOI to the datasets must be included in the Data Accessibility section of the article (<https://royalsocietypublishing.org/rsos/for-authors#question17>). Reference(s) to datasets should also be included in the reference list of the article with DOIs (where available).

Please include a Data Availability section after your main text stating where supporting data are available from, or where they will be made available should your article be accepted for publication.

If you wish to submit your supporting data or code to Dryad (<http://datadryad.org/>), or modify your current submission to dryad, please use the following link:
<http://datadryad.org/submit?journalID=RSOS&manu=RSOS-211353.R1>

- Competing interests

Please include a Competing Interests section after your main text declaring any financial or non-financial competing interests. If you have no competing interests please state 'I/we have no competing interests.'

- Authors' contributions

Please include an Authors' Contributions section at the end of your main text detailing the contribution of each author. All authors should have read and approved the manuscript before submission and this should be stated in the Authors' Contributions section.

The list of Authors should meet all of the following criteria; 1) substantial contributions to conception and design, or acquisition of data, or analysis and interpretation of data; 2) drafting the article or revising it critically for important intellectual content; and 3) final approval of the version to be published.

- Acknowledgements

- Funding statement

Please include a funding section after your main text which lists the source of funding for each author.

Because the schedule for publication is very tight, it is a condition of publication that you submit the revised version of your manuscript before 02-Dec-2021. Please note that the revision deadline will expire at 00.00am on this date. If you do not think you will be able to meet this date please let me know immediately.

Kind regards,
Dr Ellis Wilde
Publishing Editor, Journals

RSC Associate Editor
Comments to the Author:
(There are no comments.)

RSC Subject Editor
Comments to the Author:
(There are no comments.)

Reviewer comments to Author:

Reviewer: 2

Comments to the Author(s)

The authors made an effort to address adequately the reviewers' concerns. Minor point: Please format the reference list according to the journal's formatting guidelines. I am happy to recommend publication after minor revision.

Author's Response to Decision Letter for (RSOS-211353.R1)

See Appendix B.

Decision letter (RSOS-211353.R2)

Dear Dr Tedstone:

Title: Transition Metal Chalcogenide Bifunctional Catalysts for Chemical Recycling by Plastic Hydrocracking: A Single Source Precursor Approach
Manuscript ID: RSOS-211353.R2

It is a pleasure to accept your manuscript in its current form for publication in Royal Society Open Science. The chemistry content of Royal Society Open Science is published in collaboration with the Royal Society of Chemistry.

Yours sincerely,
Ellis Wilde
Publishing Editor, Journals

RSC Associate Editor
Comments to the Author:
(There are no comments.)

Reviewer(s)' Comments to Author:

Appendix A

Reviewers' Comments to Author:

Reviewer: 1

Comments to the Author(s)

The author synthesized nickel based catalyst deposited on the zeolite support followed with sulfide treatment. As prepared catalyst exhibits excellent selectivity toward gas and liquid products and has certain application prospects, it's recommended to publish in the Royal Society Open Science.

(1) The sample should be defined when it first appeared. **This has been corrected in each instance of the first appearance of each catalyst, and sample names are systematically defined in Table 1.**

(2) Why does the as-prepared catalyst have excellent selectivity? More characterizations and discussion should be provided. **More characterisation and discussion has been provided in line with this comment and those of reviewer 2.**

(3) The performance of the Pt@Beta (WI) catalyst with different proportions should be investigated. **This has been more systematically investigated in previous studies by this team of authors, including varying Si/Al ratios and other zeolite structures. The chosen zeolite structure, Si/Al ratio and Pt w% loading were informed by these previous studies as a 'state-of-the-art' high performance catalyst. These studies are cited within the current manuscript, and are also provided here for reference**

<https://doi.org/10.1016/j.micromeso.2021.110912>

<https://doi.org/10.1021/acs.iecr.9b04263>

(4) Metal chalcogenides are promising materials for the next generation of electronic devices. It not only can be used as catalysts for heterogeneous catalytic reactions, but also can be used as electrode materials for batteries and supercapacitors. I suggest authors cite or highlight some recent important progresses in this field. **We thank the reviewer for this suggestion and have included some literature highlights for interested readers.**

Reviewer: 2

Comments to the Author(s)

The present manuscript describes the synthesis and characterization of beta-supported nickel (Ni@Beta) hydrocracking and hydrotreating catalyst, as an alternative to wet impregnation using aqueous nickel (II) nitrate, to provide catalytic materials with higher conversion to fluid products in hydrocarbon refining, at the same mild batch reaction conditions of 330 °C with appropriate agitation and 20 bar H₂ pressure. The authors demonstrated that the 5wt%Ni@Beta yielded a >95 wt% conversion of a mixed polyolefin feed to fluid products even though there were coke formation issues that could be investigated further in the future. The catalysts have been characterised by, PXRD, EDX, BET, TGA and NH₃-TPD. The products have been identified by GC-MS. The preparation of the compounds is clear while the manuscript is in general well written and referenced.

The comparison of acidities hasn't been discussed very clearly. Even though the concentration of metal is a crucial factor how the acidities contribute to the performance of the catalytic process? What information we can extract from table 1? Also if would worth including the parameters in order of higher impact on the specific catalytic process. Porosity? Transition metal? Operating temperature? Acidity? The authors might want to improve this part of the discussion.

The discussion of acidity and other important factors has been extended. Whilst the aim of this study is to demonstrate the possibility of platinum alternatives, it demonstrates that the coking resistance of platinum is exceptional. It is, however, difficult to assign a hierarchy of importance to the catalyst parameters in simple terms, but we hope that the expanded discussion goes some way to helping future researchers select these properties for optimising their own experiments.

The pXRD in the supplementary information (figure 2) reads “pXRD of selected catalysts demonstrating the preservation of crystal structures.” It is not clear what the authors indicate here by “preservation”. Is it before and after the catalytic cycle? I might have missed something here but, if the authors want to show any kind of preservation, they need to compare it with something else. **The definition of preservation has been clarified in this context in the figure caption for the figure mentioned. In this case, pXRD is included to demonstrate to the reader the differing levels of structured porosity in the catalyst supports, and that activation procedures do not destroy or alter this microporous structure. Additionally, the lack of peaks in crystallography data corresponding to Ni (0) and NiS phases suggests a highly dispersed phase of Ni.**

The authors mentioned that the catalysts have been characterised by BET but there are no data in the manuscript or SI. What surface areas have been identified? How do they compare with other catalysts? **BET data has been added to the SI, and we thank the reviewer for highlighting this omission.**

This is an interesting piece of work with the main highlight in my opinion is the demonstration of the importance of preparation method of the catalyst which led to substantially higher performance without the use of noble metals even though some coking issues are present. I do believe that the manuscript can be potentially appropriate for publication in Royal Society open science but only after major revision and clarification of the points raised above.

Journal Name: Royal Society Open Science

Journal Code: RSOS

Online ISSN: [2054-5703](https://doi.org/10.1093/rsos/2054-5703)

Journal Admin Email: openscience@royalsociety.org

Journal Editor: Dr Ellis Wilde

Journal Editor Email: chemistryopenscience@rsc.org

MS Reference Number: RSOS-211353

Article Status: SUBMITTED

MS Dryad ID: RSOS-211353

MS Title: Transition Metal Chalcogenide Bifunctional Catalysts for Chemical Recycling by Plastic Hydrocracking: A Single Source Precursor Approach

MS Authors: Tedstone, Aleksander; Bin Jumah, Abdulrahman; Asuquo, Edidiong; Garforth, Arthur

Contact Author: Aleksander Tedstone

Contact Author Email: aleksander.tedstone@manchester.ac.uk,
aleksander.tedstone@manchester.ac.uk

Contact Author Address 1: University of Manchester

Contact Author Address 2: Oxford Road

Contact Author Address 3: Manchester

Contact Author City: Manchester

Contact Author State: Not US or Canada

Contact Author Country: United Kingdom of Great Britain and Northern Ireland

Contact Author ZIP/Postal Code: M13 9PL

Keywords: Recycling, Hydrocracking, Catalysis, Zeolites, Sustainability

Abstract: Sulfided nickel, an established hydrocracking and hydrotreating catalyst for hydrocarbon refining, was synthesised on porous aluminosilicate supports for the hydrocracking of mixed polyolefin waste. Zeolite Beta, zeolite 13X, MCM41 and an amorphous silica-alumina catalyst support were impregnated with the single source precursor (SSP) nickel (II) ethylxanthate for catalyst support screening. Application of this synthesis method to beta-supported nickel (Ni@Beta), as an alternative to wet impregnation using aqueous nickel (II) nitrate, provided catalytic materials

with higher conversion to fluid products at the same mild batch reaction conditions of 330 °C with appropriate agitation and 20 bar H₂ pressure. Mass balance quantification demonstrated that SSP-derived 5wt%Ni@Beta yielded a >95 wt% conversion of a mixed polyolefin feed to fluid products, compared to 39.8 wt% conversion in the case of 5wt%Ni@Beta prepared by wet impregnation. Liquid and gas products were quantitatively analysed by GC-FID and GC-MS, revealing a strong selectivity to saturated C₄ (37.3 wt%), C₅ (21.6 wt%) and C₆ (12.8 wt%) hydrocarbons in the case of the SSP-derived catalyst.

EndDryadContent

Appendix B

RSC Associate Editor

Comments to the Author:

(There are no comments.)

RSC Subject Editor

Comments to the Author:

(There are no comments.)

Reviewer comments to Author:

Reviewer: 2

Comments to the Author(s)

The authors made an effort to address adequately the reviewers' concerns. Minor point: Please format the reference list according to the journal's formatting guidelines. I am happy to recommend publication after minor revision.

The manuscript has had the reference section reformatted to the journals formatting guidelines.

Modification of the Ethics Statement and Data Repository availability has also been performed in line with journal guidelines.